# Identification of malignant cells in single-cell transcriptomics data
Massimo Andreatta [1,2,3] ✉, Josep Garnica [1,2,3] & Santiago Javier Carmona[1,2,3]

Single-cell transcriptomics has significantly advanced our ability to uncover the cellular heterogeneity of tumors. A key challenge in single-cell transcriptomics is identifying cancer cells and, in particular, distinguishing them from non-malignant cells of the same cell lineage. Focusing on features that can be measured by single-cell transcriptomics, this review explores the molecular aberrations of cancer cells and their observable readouts at the RNA level. Identification of bona fide cancer cells typically relies on three main features, alone or in combination: i) expression of cell-of-origin marker genes; ii) inter-patient tumor heterogeneity; iii) inferred copy-number alterations. Depending on the cancer type, however, alternative or additional features may be necessary for accurate classification, such as single-nucleotide mutations, gene fusions, increased cell proliferation, and altered activation of signaling pathways. We summarize computational approaches commonly applied in single-cell analysis of tumoral samples, as well as less explored features that may aid the identification of malignant cells.

Solid tumors are complex ecosystems consisting of multiple interacting populations. Besides malignant cells, a heterogeneous collection of non-malignant immune and stromal cells populate the tumor microenvironment (TME), and affect the ability of the tumor to grow, invade other tissues, and resist therapies. Both malignant and non-malignant cells in the TME can take up multiple cellular states, depending on many factors including tissue location, nutrient availability, cell-cell interactions, and inflammatory conditions[1,2]. Single-cell omics technologies, and in particular single-cell RNA-sequencing (scRNA-seq), have revolutionized our ability to describe with unprecedented resolution the heterogeneity of cancer cells and of their TME[3].

Cell type annotation is a key step towards understanding the diversity of scRNA-seq datasets. Cell types can be manually assigned by clustering and expression of marker genes, or by applying one of many automated computational tools[4]. In the context of cancer, a crucial aspect is the identification of malignant cells from complex single-cell data sets composed of diverse immune and stromal cell types, as well as non-malignant cells of the same lineage as the cancer cells (e.g., normal epithelial cells). Distinguishing malignant cells from their healthy counterparts at the single-cell level is especially challenging – in particular in primary tumors where both malignant and normal cells of the same lineage coexist. While tumors are inherently unique – even within the same cancer type – a limited number of underlying organizing principles (or "hallmarks") govern the acquisition of the traits that lead to malignancy[5,6]. With a pragmatic focus on measurable features in scRNA-seq, we will explore the molecular alterations and idiosyncrasies of cancer cells, along with their observable transcriptional phenotypes (Fig. 1). In this context, we will review the computational approaches commonly applied in research practice as well as additional features that may enhance the identification of malignant cells from single-cell transcriptomics data.

## Expression of cell-of-origin markers

The "cell of origin" (COO) refers to the normal cell type that underwent malignant transformation and gave rise to the tumor[7]. For instance, carcinomas originate from epithelial cells; sarcomas from mesenchymal cells; lymphomas and leukemias from hematopoietic cells. In the context of malignant cell identification in scRNA-seq data, one of the most straightforward approaches to isolate the cell type of interest is the use of COO markers. By way of example, Puram et al. applied a signature of epithelial genes to identify malignant cells in head-and-neck squamous cell carcinoma (HNSCC)[8]. They found that expression of epithelial genes successfully partitioned putative cancer cells from immune and stromal components, and that this classification agreed with orthogonal readouts such as copy-number alterations. In their meta-analysis across nine cancer types, Barkley et al. relied on the expression of COO markers (epithelial markers for carcinomas, stromal markers for gastrointestinal stromal tumors) as one of the criteria to identify malignant cells[9]. In multiple myeloma (MM), a signature of plasma cell markers (*MZB1, JCHAIN, SDC1*) was sufficient to isolate malignant MM cells[10]; the authors then confirmed that these cells harbored copy-number

[1]Department of Pathology and Immunology, Faculty of Medicine, University of Geneva, 1206 Geneva, Switzerland. [2]Swiss Institute of Bioinformatics, 1015 Lausanne, Switzerland. [3]Translational Research Centre in Onco-Hematology (CRTOH), Geneva, Switzerland. ✉e-mail: massimo.andreatta@unige.ch

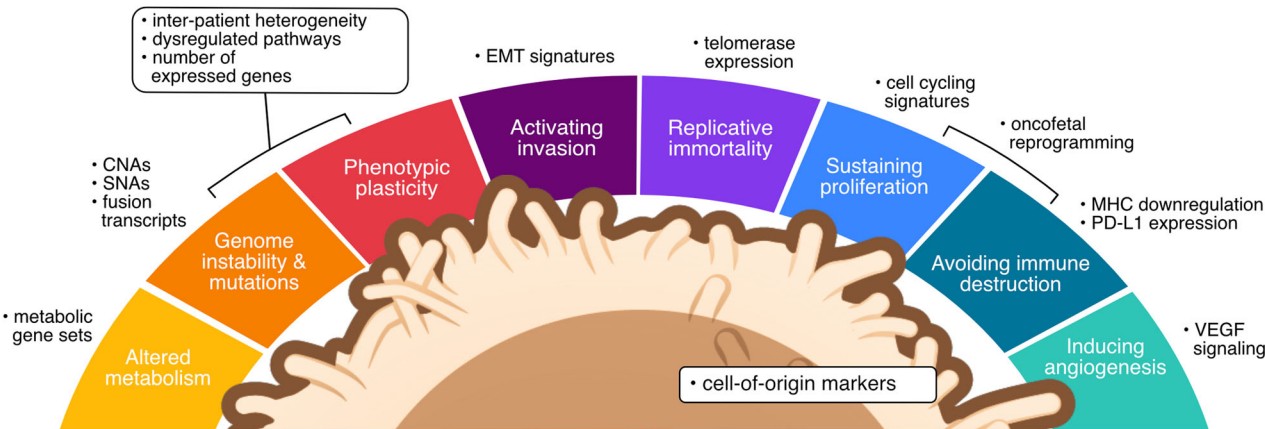

**Fig. 1 | Transcriptional aberrations that can be quantified by scRNA-seq, grouped by cancer hallmarks.** In bold are indicated the features most frequently used in scRNA-seq analysis to identify malignant cells. Cell illustration modified from NIAID NIH BIOART (bioart.niaid.nih.gov/bioart/508).

alterations that were not present in naïve and memory B cell populations, or in normal plasma cells from a different study.

However, tumors often contain – in addition to cancer cells – normal cells from the same COO lineage. A recent meta-analysis by Gavish et al. estimated that two-thirds of scRNA-seq carcinoma samples contained a variable fraction of non-malignant epithelial cells[11]. Therefore, expression of COO markers alone is not sufficient to distinguish malignant cells and should be complemented by other features (see Table 1). In their analysis of nasopharyngeal carcinoma, Chen et al. first distinguished cells of epithelial origin (both malignant and normal) from immune and stromal cells by expression of epithelial gene sets. Secondly, they subset on the epithelial compartment and predicted copy-number alterations to distinguish normal from malignant epithelial cells[12]. Similarly, Kürten et al. identified epithelial cells in HNSCC by unsupervised clustering and marker-based annotation, and then separated normal from malignant epithelial cells by their copy-number profiles[13]. In infantile fibrosarcoma (a soft-tissue malignancy that originates from fibroblasts), Li et al. utilized specific markers (*MMP9*, *COL1A1*) to focus on fibroblasts, and subsequently applied InferCNV to distinguish malignant from non-malignant cells[14]. A potential complication with using COO markers in several carcinomas is represented by epithelial-to-mesenchymal transition (EMT), whereby cancer cells lose or down-regulate expression of epithelial markers such as *EPCAM* and *CDH1* while gaining mesenchymal markers such as *VIM* and *FN1*, giving rise to inter-mediate differentiation states that may be difficult to detect based only on markers[15,16]. On the other hand, expression of EMT markers could poten-tially be exploited to distinguish malignant from normal epithelial cells (see the section "Activating invasion" below). In summary, cell-of-origin mar-kers are generally effective for distinguishing tumor cells from stromal and immune cells but need to be complemented by other readouts to differ-entiate malignant from non-malignant cells of the same lineage.

## Copy number alterations

A copy-number alteration (CNA) refers to the aberrant duplication or deletion of genomic DNA segments. Some literature uses CNA inter-changeably with CNV (copy-number variation), although the latter usually refers more specifically to the germline variation between individuals in a population. By amplifying the expression of oncogenes or dampening the effect of tumor suppressor genes, CNAs have a crucial role in cancer development[17,18]. CNAs and aneuploidy – the loss or duplication of entire chromosomes– are extremely common in tumor formation and progres-sion: it has been estimated that approximately 90% of solid tumors and 75% of hematopoietic cancers are aneuploid[19,20]. Moreover, tumors of similar origin tend to have similar chromosomal aberrations. For example, squa-mous cancers tend to lose chromosome arm 3p[21]. Gastrointestinal tumors of different types (colorectal, non-squamous esophageal, stomach, and pan-creatic) present frequent co-occurring gains of arms 8q, 13q, and

chromosome 20[21]. In a multiple myeloma cohort, 17 of 21 patients harbored a deletion of chromosome 13[22]. It should be noted that CNAs and aneu-ploidy occur at different scales (specific regions versus whole chromosomes) and following different mechanisms (local duplication/deletion versus defective chromosomal segregation). However, from the point of view of their detection in the transcriptome, both processes are measured as an increase/decrease of average expression over a chromosomal region, and the same computational methods are commonly applied for their detection. For the sake of simplicity, here we will refer to both processes as copy-number alterations.

Several computational methods have been developed for the pre-diction of CNAs in scRNA-seq data. InferCNV is one of the first and most widely used CNA prediction methods to date[23]. Briefly, the algo-rithm calculates the smoothed expression of genes ordered along their chromosomal coordinates, and compares this profile to the corre-sponding expression in a population of diploid "reference" cells (which may be normal cells of the same lineage, or other "confident normal" cells such as immune cells, as discussed later in this section). CNA events are predicted using a hidden Markov model that evaluates the transition to complete or partial loss/duplication of chromosome regions, with an optional refinement by a Bayesian mixture model. CopyKAT[24] combines hierarchical clustering with a Gaussian mixture model to identify a population of "confident normal" cells, which is used to estimate copy number baseline values for diploid cells. Based on this reference, chro-mosomal breakpoints are calculated through a statistical framework that tests for significant mean expression differences between adjacent genomic windows. Similar to CopyKAT, SCEVAN[25] starts by identifying a small set of confident normal cells and the diploid baseline; it then uses a joint segmentation algorithm to identify breakpoints and deviations from the baseline. In addition to the gene expression profile along the chromosomes, CaSpER[26] measures an allelic shift signal to estimate genome-wide loss-of-heterozygosity events. The measurement of allelic shift requires calculating single-nucleotide variants (SNVs); therefore, this method must be run on the sequencing reads (unlike most other approaches, which can be applied directly on the gene expression matrix). Numbat[27] complements gene expression profiles with haplotype information and allelic imbalance estimates to support CNA calls. Recent benchmarks have found that methods that exploit allelic shift signals (such as Numbat and CaSpER) have superior performance for CNA identification; when only expression matrices are available, CopyKAT is the recommended method[28,29].

Regardless of the algorithm employed, information in single-cells is considered to be too noisy for classification. Instead, cells are usually first clustered based on the CNA global patterns, and then all cells in a cluster are collectively classified as either normal or malignant[8,23]. Identification of malignant clusters can be supported by previous knowledge on common

**Table 1 | Transcriptional features commonly or potentially used to identify malignant cells from scRNA-seq data**

| Commonly used features | | |
|---|---|---|
| **Feature/aberration** | **Readout** | **Comments** |
| Expression of cell-of-origin-marker genes | Gene signature score | Not sufficient to distinguish between normal and malignant cells of the same type; usually combined with other features |
| Inter-patient tumor heterogeneity | Index of cluster mixing (e.g. LISI score, entropy) | Requires multiple samples; may be confounded by batch effects |
| Copy-number alterations | Copy-number profile/aneuploidy score | Requires a reference of "normal" ploidy; will not detect malignant cells without chromosomal alterations |
| Supporting features | | |
| **Feature/aberration** | **Readout** | **Comments** |
| Single-nucleotide alterations and mutational burden | Mutations in known sites/total number of mutations | Works best when combined with WES of matched samples; limited by low-coverage of scRNA-seq technologies |
| Formation of fusion transcripts | Expression of fused genes | Specific to individual cancer types; limited by low-coverage of scRNA-seq technologies |
| Sustained proliferation | Signature score for cycling gene sets | Commonly measured as cycling enrichment by cluster |
| Pathway dysregulation | Signature score for altered pathway | Specific to individual cancer types |
| Potentially discriminating features | | |
| **Feature/aberration** | **Readout** | **Comments** |
| MHC downregulation | Signature score for antigen-presenting machinery | Specific to individual cancer types, TMEs, or individual cancer sub-clones |
| Overexpression of checkpoint molecules | Checkpoint ligand expression | Limited evidence in scRNA-seq |
| Expression of telomerase subunits | Gene or signature score | Limited evidence in scRNA-seq |
| Metabolic signatures | Signature score | Adjacent normal cells may exhibit similar alterations |
| Pro-angiogenic signaling | Gene or signature score | Limited evidence in scRNA-seq |
| Drivers of invasion (EMT) | Signature score | Intermediate EMT states may be difficult to capture |
| Oncofetal reprogramming | Gene or signature score | Specific to individual cancer types |
| Number of unique expressed genes | Gene count | Can be confounded by heterogeneous sequencing depth |

chromosomal alterations for the cancer type in question. For example, a study of clear cell renal cell carcinoma (ccRCC) classified as malignant all scRNA-seq clusters with chromosome 3p loss, a common alteration in ccRCC[30]. When available, paired whole-exome sequencing (WES) can also be useful to support the major CNAs predicted by scRNA-seq. For instance, Xing et al. used ploidy alterations derived from WES to confirm the chromosomal regions most affected by CNAs in multiple lung adenocarcinoma patients[31]. Chen et al. corroborated CNA patterns obtained in scRNA-seq by matched WES on the same patients[12].

To support the identification of malignant cells harboring CNAs, it can be advantageous to have access to bona fide non-cancerous samples as references for normal ploidy. In a study of nasopharyngeal cancer[12], the authors also collected one sample from normal nasopharyngeal epithelial tissue. They applied InferCNV to all cells expressing epithelial markers in both cancer and normal control samples. Cells from tumor samples with CNA profiles that clustered with the normal sample rather than the tumor sample were considered as normal epithelial cells. Maynard et al. performed scRNA-seq on 45 samples of lung adenocarcinoma, as well as on normal adjacent tissue. They found that the CNA profile of a group of epithelial cells – classified as non-malignant– clustered with spike-in normal cells from adjacent tissues, while cells believed to be malignant formed a separate cluster compared to the spike-in normal cells[32]. When no normal samples are available as a reference, immune cells are often used as a baseline for normal ploidy. For instance, a recent study in Non-Small Cell Lung Cancer (NSCLC) considered dendritic cells, which were abundant across all patients, as the confident diploid reference for CNA detection by CopyKAT[33].

Kim et al. proposed a quantitative approach to automatically label cells as normal or malignant based on CNA profiles: first they measured the sum of squared values for the CNA profiles, calculated as altered expression on moving windows of 100 genes, compared to a normal reference; based on

the top 5% of cells along this axis (assumed to be robustly malignant), they calculated the correlation of the CNA of each cell with the profile of these robustly malignant cells. High-correlating cells, defined by a fixed threshold across samples, were labeled as malignant[34]. The predictive value of this strategy was independently confirmed in a meta-analysis of multiple cancer types[9]. In specific cancer types, such as thyroid cancer, pediatric cancers, sarcomas, and certain hematopoietic cancers, CNAs are more rare, and other mechanisms such as gene translocation, point mutations and epigenetic changes drive carcinogenesis[35–38]. In these cases, identification of malignant cells in scRNA-seq should rely on additional transcriptional features.

## Inter-patient tumor heterogeneity

Inter-patient tumor heterogeneity refers to transcriptional differences between cancer cells of different patients. Because of their distinct mutational histories, spatial location and interactions with the environment, different tumors are in general genotypically and phenotypically unique. Variability among cancers of the same organ is also at the basis of classification of cancers into subtypes, usually characterized by distinct morphologies, oncogenic drivers and expression of specific markers[7].

In exploratory single-cell transcriptomics analysis, inter-patient variation tends to be a dominant factor and often results in patient-specific cell clusters. Several studies have observed such patient specificity and leveraged it to distinguish malignant cells from normal cells. For example, Tirosh et al. showed that malignant cells (as orthogonally defined by ploidy alterations) from melanoma tumors tended to form individual patient-specific clusters[39]. On a large ovarian cancer cohort, Vázquez-Garcia et al. observed that, whereas stromal and immune cells from different patients tended to cluster by cell type, the transcriptomes of cancer cells were patient-specific[40]. Bischoff et al. collected multiple lung adenocarcinoma samples, as well as normal lung tissue; upon performing scRNA-seq, cells that formed

**Table 2 | Selected tools and resources for the identification of malignant cells in scRNA-seq data**

| Resource | Type/readout | Comments | Availability and references |
|---|---|---|---|
| InferCNV | Copy number alterations | Arguably the most widely used method for CNA detection in scRNA-seq | https://github.com/broadinstitute/infercnv[23] |
| CopyKAT | | Among top performers in recent benchmarks, especially when using only gene expression matrix | https://github.com/navinlabcode/copykat[24] |
| Numbat | | Exploits allelic imbalance to improve CNA prediction; requires sequencing reads | https://github.com/kharchenkolab/numbat[27] |
| LISI | Inter-patient heterogeneity | A simple metric of patient mixing | https://github.com/immunogenomics/LISI[45] |
| scIntegrationMetrics | | Implements per-cell-type LISI and additional metrics | https://github.com/carmonalab/scIntegrationMetrics[129] |
| scAllele | Single nucleotide alterations | SNA detection tailored for scRNA-seq | https://github.com/gxiaolab/scAllele[50] |
| Monopogen | | SNA calling (germline + somatic) leveraging linkage disequilibrium from reference panels | https://github.com/KChen-lab/Monopogen[51] |
| STAR-fusion | Fusion transcripts | Primarily designed for bulk RNA-seq, but can be adapted for single-cell data | https://github.com/STAR-Fusion/STAR-Fusion[62] |
| scFusion | | Specific for gene fusion detection at single-cell resolution | https://github.com/XiDsLab/scFusion[65] |
| UCell | Gene signature scoring | Simple and robust rank-based gene set scoring | https://github.com/carmonalab/UCell[130] |
| GSVA | | Implements methods for gene set enrichment analysis | https://github.com/rcastelo/GSVA[131] |
| scATOMIC | Automated classifier | Integrated pipeline for cell type classification, including malignant vs. normal cells | https://github.com/copykat-lab/scATOMIC[82] |
| Ikarus | | Relies on DEG signatures between normal and malignant cells | https://github.com/BIMSBbioinfo/ikarus[122] |
| scMalignantFinder | | Uses logistic regression trained on curated pan-cancer gene signatures and DEGs | https://github.com/Jonyyqn/scMalignantFinder[123] |
| OncoDB | Database | Collates expression profiles for cancer vs. normal tissues | https://oncodb.org/[81] |
| 3CA | | Provides robust transcriptional meta-programs for several cancer types | https://www.weizmann.ac.il/sites/3CA/[114] |
| HPA | | Includes scRNA-seq expression profiles for many tissues and cell types | https://www.proteinatlas.org/[132] |

patient-specific clusters were labeled as malignant, whereas cells that clustered together with the normal lung tissue were labeled as normal epithelial cells[41]. Inter-patient heterogeneity has also been observed in non-small-cell lung cancer[42], HNSCC[8], as well as in hematopoietic cancers. For instance, a study of childhood leukemia found that malignant cells formed patient-specific clusters, whereas normal bone-marrow mononuclear (BMMC) cells from leukemia samples clustered with BMMCs from healthy donors[43].

Aiming at quantifying inter-patient heterogeneity in basal cell carcinoma (BCC), Yerly et al.[44] measured cluster mixing by a per-cell-type Local Inverse Simpson's Index (LISI) score[45]. For a given cell type, this metric quantifies the number of patients having cells in any given neighborhood of cells, effectively measuring patient mixing for each cell type (see resources in Table 2). Across a cohort of BCC patients, cancer cells had consistently lower LISI scores compared to normal epithelial cells, stromal cells, and all immune cell types[44]. Similarly, Chan et al. observed that malignant cells had lower mixing (measured in terms of Shannon entropy) compared to normal epithelial cells as well as to non-epithelial cells[46]. Batch effects – technical variations introduced during sample collection and processing, library preparation and sequencing – can be a potential confounding factor when assessing inter-patient heterogeneity. Therefore, the evaluation of inter-patient tumor heterogeneity and its associated metrics depend on the availability of multiple samples from different individuals sequenced in a consistent way (i.e. same technology and protocol). Considering the scale and experimental design of modern scRNA-seq experiments, this should not be a frequent limitation.

## Single-nucleotide alterations and mutational burden

While normal tissues tend to accumulate somatic mutations over time, cancer cells typically harbor a higher number of mutations compared to their normal counterparts[47]. Therefore, single-nucleotide alterations (SNAs) in specific driver genes, as well as the total number of mutations accumulated by a given cell (the "mutational burden"), have the potential to discriminate normal from malignant cells. However, SNAs are challenging to measure from scRNA-seq data. Most scRNA-seq technologies produce relatively short reads, which are usually enriched at 3' ends; sequencing coverage is often low and uneven along the genome; and RNA profiles can be affected by allelic imbalance[48]. A benchmark of SNA detection algorithms showed that most traditional tools for bulk analysis detected a very low fraction of variants in scRNA-seq data[49]. More recent tools specifically developed for single-cell omics data, such as scAllele, Monopogen, and SComatic, perform better[50–52], but their application remains mostly limited to high-coverage single-cell sequencing technologies[48].

Gasper et al. demonstrated that variant calling by SnpEff[53] allowed the identification of cancer cells on several Smart-seq2 datasets that had otherwise limited or no copy-number variations[54]. However, they also remarked that their approach was likely to fail on 10X Genomics datasets, due to the reads covering only a short region of the 3' end of each transcript. In a small cell lung cancer (SCLC) study, Chen et al. predicted SNAs on scRNA-seq data, and then classified cell clusters to be malignant if they were enriched in reads calling SNAs compared to immune and mesenchymal cells[46]. Ianevski et al. implemented a single-cell SNA calling module based on transcriptomics data in their scType tool, and showed that the number of SNA within a cell type was predictive of malignant vs. normal cells for acute myeloid leukemia (AML)[55].

The potential of SNA detection from scRNA-seq will likely be unlocked by technological advances in sequencing methods. Aiming at extending the range of detectable mutations, van Galen et al. combined well-based scRNA-seq with targeted DNA sequencing on a panel of common mutations in AML, a cancer type that frequently lacks CNAs[38]. They also incorporated long-read nanopore sequencing in their pipeline, allowing improved detection of mutations far from 3' gene ends. Single-cell mutation calls and transcriptomes were used to train a machine learning classifier with the ability to distinguish normal from malignant bone marrow cells. By combining full-length transcriptome coverage with a 5' unique molecular

identifier (UMI) RNA counting strategy, the Smart-seq3 technology enables the identification of SNAs at allele and isoform resolution, making Smart-seq3 particularly suitable for distinguishing malignant cells from normal cells in heterogeneous tumor samples[56]. While specific studies employing Smart-seq3 for SNA detection in cancer are yet to be published, this technology inherently supports such applications.

## Formation of fusion transcripts

Gene fusions resulting from chromosomal rearrangements have been observed at high frequencies in specific cancer types, including sarcomas, leukemias, and prostate cancer[57–59]. Fusion transcripts can generate onco-proteins, as in the case of the TMPRSS2–ERG fusion after chromosome 21 rearrangement in prostate cancer[60], the EWSR1–FLI1 chimeric oncogene observed in 85% of Ewing sarcomas[37], or the long-recognized BCR-ABL1 fusion observed in the "Philadelphia chromosome" of chronic myeloid leukemia patients[61].

Multiple computational tools have been developed for the detection of fusion transcripts in bulk RNA-seq data, such as STAR-fusion[62] and Arriba[63]. When aligning RNA-seq reads to a reference genome, these methods focus on the detection of chimeric (fusion) transcripts, where a single read maps to two distinct gene loci. Several parameters – such as the number of supporting reads, breakpoint consistency, intergenic distance, and presence in databases of known fusions – are often applied to prioritize and filter candidate fusion events. While developed for bulk transcriptomes, these methods have also been adapted to high-coverage scRNA-seq data. For example, Jerby-Arnon et al. applied STAR-fusion to predict SS18–SSX fusion transcripts from a Smart-Seq2 dataset of synovial sarcoma, and observed that a large fraction of cancer cells harbored this aberration[64]. Computational tools specifically developed for scRNA-seq such as scFusion are emerging[65], but their usefulness remains to be benchmarked in low-coverage single-cell datasets.

## Sustained proliferation

The ability to sustain chronic proliferation is one of the fundamental hallmarks of cancer[5]. While in normal cells growth and cell cycling signals are tightly regulated, ensuring homeostatic cell numbers and tissue architecture, cancer cells acquire the ability to grow and multiply uncontrollably. In single-cell data, meta-analyses have revealed that cell cycling is one of the most conserved transcriptional programs in cancer cells[11,66]. However, because scRNA-seq captures a static snapshot, only a fraction of cells will be actively progressing through the cell cycle (outside of G0) at any given time. In a fast-proliferating cancer such as HNSCC, Puram et al. observed that between 14% to 40% of malignant cells were cycling, depending on the individual tumor[8]. A melanoma study found high variability in the proliferation status of tumors, with 13.5% of cells cycling on average across individuals[39]. In other kinds of tumors, such as low-grade glioma, proliferating cells are considerably less frequent[67].

While only a fraction of cancer cells are expected to be cycling in scRNA-seq datasets, the rate of proliferating cells within a cell cluster can be useful to distinguish malignant cells from normal cells. In a study of pancreatic adenocarcinoma, Lin et al. evaluated signatures of cell cycling phases and found that clusters of cancer cells were more proliferative than normal epithelial cells or fibroblasts[68]. A study of human and mouse cancers defined cancer cells (among epithelial cells) as patient-specific clusters with an enrichment of Mki67 expression[42]. In cervical squamous cell carcinoma (CSCC), expression of cell cycling genes was significantly increased compared to epithelial cells from the normal cervix[69]. While cell proliferation is quantified primarily in terms of gene-set scoring for cell cycling phases, machine-learning methods trained specifically for the task of cell cycle-phase prediction have also been proposed (reviewed in ref. 70). Machine-learning methods have the ability to learn complex, multivariate relationships between genes and may provide more robust cell cycle-phase predictions. Because snapshot scRNA-seq data can capture the proliferation rate only within a group of cells and not for single cells, it is mainly used as a confirmatory readout rather than a criterion for malignant cell classification.

## Dysregulation of signaling pathways

Cancer exploits existing signaling pathways to grow, evade immune detection, and survive in a hostile environment. Critical molecular pathways frequently altered in cancer include the PI3K/AKT/mTOR pathway, the RTK/RAS pathway, and the TP53 pathway (see e.g. refs. 71,72). Other alterations are more specific to individual cancer types. For example, dys-regulation of the androgen receptor pathway plays a crucial role in the development and progression of cancers of the prostate, the most androgen-responsive tissue[73]. Mutations in components of the hedgehog pathway lead to hyperactivation of HH signaling in basal cell carcinoma and medulloblastoma[74].

These aberrant pathways are reflected in altered transcriptomes, with specific genes that can be used as biomarkers for a given cancer type. By way of example, HE4 (coded by WFDC2) has been consistently observed as overexpressed in ovarian carcinomas[75]; TM4SF1 and LAMC2 are over-expressed in pancreatic ductal adenocarcinoma[76,77]; other gastrointestinal cancers frequently harbor activating mutations and overexpression of KIT and PDGFRA[78]. Elevated expression of the MYC oncogene has been reported in up to 70% of cancers[79]. Zhang et al. calculated a "malignant score" for single cells in gastric adenocarcinoma based on a signature of differentially expressed genes between matched tumour and normal TCGA samples (including CLDN4, CLDN7, TFF3, and REG4); they then used this score to distinguish malignant from normal epithelial cells[80]. The OncoDB database collects molecular signatures of differentially expressed genes between cancer cells and normal cells, derived from bulk transcriptomes for multiple cancer types[81]. scATOMIC, a method for pan-cancer cell type classification, leverages OncoDB signatures to distinguish malignant from normal cells based on signature scoring and hierarchical clustering[82].

In a study of basal cell carcinoma (BCC), Yerly et al. took advantage of the aberrant expression of several genes involved in the HH pathway (including PTCH1, HHIP, GLI1, and GLI2) to identify malignant cells in scRNA-seq data[44]. By constructing a signature from this gene set, they observed a higher signature score in cancer cells compared to normal ker-atinocytes (the cell of origin for this cancer type). The signature-based classification was largely in agreement with orthogonal criteria for malig-nant cell identification, namely inter-patient tumor heterogeneity and CNA patterns[44]. The authors were also able to confirm the predictive power of such a signature to isolate malignant cells in an independent, previously published BCC cohort[83].

Upon performing scRNA-seq in a pancreatic cancer cohort, Peng et al. observed two distinct groups of ductal cells; they inferred that one group represented malignant ductal cells based on both altered CNA profiles and upregulation of several programs such as cell adhesion, response to stress, and cell proliferation[84]. To distinguish malignant from non-malignant B cells, Roider et al. exploited the fact that malignant B-cell populations tend to express only one type of immunoglobulin light chain (either a kappa or lambda light chain)[85]. By calculating the expression ratio of the constant part of the kappa and lambda chains (defined as $IGKC / (IGKC + IGCL2)$), the authors were able to classify B-cell clusters either as healthy (approximately 0.5 ratio) or malignant (ratio skewed towards 0 or 1).

In summary, many cancer types co-opt developmental plasticity to increase their proliferation, invasiveness, and therapy resistance. However, different cancer types disrupt distinct molecular pathways depending on the tissue of origin and common genetic alterations. For this reason, the use-fulness of gene signatures associated with such dysregulated pathways for the identification of cancer cells is in most cases cancer-type specific.

## Additional transcriptional features

Defective MHC presentation. Tumor cells can evade CD8 T cell immune detection by developing deficiencies in the MHC class I antigen presentation pathway. Multiple mechanisms can be used by cancer cells to reduce antigen presentation on their surface, including genetic and

epigenetic changes, as well as antigen depletion[86]. Genetic changes can affect the core components of the MHC complex (the variant HLA-A/B/C chains and the invariant B2M chain), as well as other components of the antigen-presentation machinery[87,88]. Non-genetic mechanisms for MHC-I downregulation include epigenetic, transcriptional and post-transcriptional silencing, as well as modulation of MHC-I expression by disruptions in interferon pathways[88]. In practice, all of these mechanisms can result in lower expression of the antigen presentation machinery, *B2M* and genes in the MHC locus. While down-regulated expression of *B2M* and MHC-I molecules has been reported in cancer cells in bulk RNA-seq studies[89], transcriptional aberrations in MHC presentation components have not been systematically investigated in scRNA-seq data. This is a largely unexplored and potentially useful readout to distinguish malignant from normal cells from scRNA-seq data.

**Overexpression of checkpoint molecules.** T cell receptors (TCR) expressed on the surface of T cells can recognize mutated or overexpressed self-antigens displayed by MHC molecules. However, multiple co-stimulatory and co-inhibitory receptors (known as immune checkpoints) modulate the TCR signaling pathway triggered in T cells, ultimately shaping the expansion, differentiation, and phenotype of tumor-reacting T cells[90]. These rheostatic mechanisms allow the detection of aberrant cells while maintaining self-tolerance and limiting healthy tissue damage. Cancer cells are known to exploit immune checkpoints to escape immune-mediated destruction[91]. The PD-1/PD-L1 pathway is arguably the most well-studied immune checkpoint in human cancers and has been the target of multiple immune checkpoint blockade therapeutic avenues. Binding of PD-L1 on the surface of cancer cells with PD-1 expressed on tumor-infiltrating T cells modulates the TCR pathway and leads to inhibition of T cell activity. Indeed, high PD-L1 expression is often observed on cancer cells and is generally associated with poor prognosis[92] – though it may prove beneficial in the context of anti-PD1 therapy[93]. The immune checkpoint molecule TIM-3, expressed in activated T cells, interacts with multiple ligands, such as Gal-9, CEACAM1, PtdSer, and HMGB1[91]. Gal-3 (encoded by *LGALS3*) is expressed in a variety of cancer cells, and it can interact with the immune checkpoint molecule LAG-3 to inhibit cytotoxic activity in CD8 + T lymphocytes[94]. Yakubovich et al. showed that expression of *LGALS3* (as well as *LGALS3BP*) correlated with epithelial-to-mesenchymal transition in ovarian cancer and is associated with worse prognosis[95]. Another LAG-3 functional ligand, FGL1, was shown to be highly expressed in a variety of cancers, such as melanoma, lung cancer, and colorectal cancer[96]. Thus, expression of co-inhibitor receptors is a promising criterion for cancer cell detection. Systematic evaluations on the detectable expression of these molecules by cancer cells in scRNA-seq data are still lacking.

**Expression of telomerase subunits.** Active telomerase is essential for cancer cells to maintain telomere length and enable replicative immortality. While telomerase can be occasionally active in normal cells and has been correlated with cell proliferation[97], upregulation of telomerase is a common mechanism of cancer and has been detected in most cancer types[98]. In particular, the transcription of telomerase reverse transcriptase (TERT), the catalytic subunit of the telomerase enzymatic complex, is tightly regulated in normal cells but is disrupted by multiple possible mechanisms in cancer cells[99]. Noureen et al. noted that the enzymatic activity of telomerase correlates only partially with the expression of TERT; instead, they suggested a 13-gene signature as a robust tool to infer telomerase activity across multiple cancer types[100].

**Metabolic reprogramming.** Cancer cells must adapt their metabolism to sustain their rapid growth and meet increasing energy demands. A well-known metabolic shift is the Warburg effect, where cancer cells preferentially utilize glycolysis for energy production instead of oxidative phosphorylation[101]. Beyond increased glycolysis, cancer cells tend to preferentially utilize specific amino acids as a major source of energy, especially glutamine, and reprogram their lipid metabolism[102,103].

Metabolic reprogramming can be quantified in terms of gene-set activity and pathway enrichment, or by using more complex constraint-based or kinetic modeling for the reconstruction of metabolic fluxes (reviewed in ref. 104). Regardless of the algorithm employed, differences in metabolic activity may be exploited to distinguish normal from malignant cells in scRNA-seq. For example, a study of lung cancer distinguished lung cancer cells from normal epithelial cells by CNA patterns and used these subsets to derive signatures of differentially expressed metabolic genes. In particular, they found that lipid metabolism was broadly dysregulated in lung cancer cells compared to normal epithelial cells[105]. A survey of the metabolic alterations in melanoma and HNSCC from scRNA-seq data found that the activities of glycolysis and oxidative phosphorylation correlated with hypoxia at the single-cell level, and that in general metabolic pathways in malignant cells were more active and plastic than those in non-malignant cells[106]. A recent study showed that, in multiple cancer types, malignant cells expressed a higher fraction of mitochondrial RNA compared to healthy cells, independently of technical artifact related to cell dissociation[107]. It remains unclear whether normal cells from the same origin, especially when they experience the same microenvironment as the cancer cells, would also adapt their metabolism in similar ways.

**Pro-angiogenic signaling.** Tumor vascularization is essential to provide nutrients and oxygen to cancer cells, as well as for disposing of metabolic waste. During cancer progression, pro-angiogenic signaling is activated to ensure access to the blood circulation system and allow tumor growth[108]. In particular, the VEGF signaling pathway has been shown to promote vascularization across many cancer types, and the VEGF-A gene is produced in large quantities by cancer cells[109]. Upregulation of fibroblast growth factors (FGF) by cancer cells has also been shown to promote angiogenesis and tumor growth[110]. In single-cell data, several studies have shown that VEGF is highly expressed by at least a subset of cancer cells[111,112]. However, to the best of our knowledge no studies have directly compared pro-angiogenic pathway activities in normal versus cancer cells from the same tissue at single cell level.

**Drivers of invasion.** Cancer invasion refers to the ability of tumor cells to break through their local tissue boundaries, penetrate the surrounding extracellular matrix, and invade nearby tissues. Epithelial-to-mesenchymal transition (EMT) is a fundamental process by which cancer cells lose their adhesion and polarity, adopting mesenchymal traits that increase their migratory and invasive capacities. These morphological and functional changes are accompanied by the expression of transcriptional programs characterized by gradual loss of epithelial markers (e.g. *EPCAM* and *CDH1*) while gaining mesenchymal characteristics (e.g. *VIM* and *FN1*)[113]. Large meta-analyses of scRNA-seq have identified recurrent EMT programs across multiple cancer types, providing signatures that could aid the identification of malignant cells[11,114]. While EMT can be a useful feature to distinguish malignant from normal cells, it has limitations: EMT is a continuous, dynamic and reversible process, which gives rise to partial and intermediate states that affect only a fraction of cancer cells in a tumor.

**Oncofetal reprogramming.** Re-expression of fetal-like cell states allows cancer cells to mimic characteristics of embryonic or fetal development, adopting gene expression patterns and behaviors typical of early developmental stages. By reactivating developmental signaling pathways and cellular states that are dormant or inactive in normal adult tissues, oncofetal reprogramming promotes tumor development and invasion, phenotypic plasticity and evasion of immune recognition[115]. Abnormal expression of fetal antigens in tumors has been reported in several cancer types and has been used as a clinical biomarker. For example, re-expression of the oncofetal proteins *SALL4* and *AFP* is often observed in hepatocellular carcinoma[116,117]. High expression of *IGF2BP* family proteins has been reported across cancers of multiple tissues, including

colon, liver, kidney, pancreas, and female reproductive organs[118]. Other common genes with oncofetal properties include *CEA*, *CA125*, *LIN28B*, *H19* and *FOXM1*[115]. Using scRNA-seq data, a recent study of cervical cancer found that oncofetal protein *SALL4* was significantly upregulated in cancerous cells compared to normal epithelial cells[119]. Because of their specific expression in malignant cells, oncofetal signatures can be potentially useful to identify cancer cells in single-cell data.

**Number of expressed genes**. A study of NSCLC reported that cancer cells expressed a significantly higher number of unique genes compared to normal epithelial cells, and that this observation could not simply be explained by sequencing depth[32]. This observation agrees with the census by Zhang et al., which found a large increase in the number of unique genes in cancer, especially for lowly-expressed genes[120]. There are several possible reasons for this effect: aberrant epigenetic regulation may activate otherwise silent genes or pseudo-genes; hyperactivation of signaling pathways such as PI3K/AKT/mTOR or RTK/RAS may lead to expression of genes that would otherwise be unutilized by normal cells; increased intra-tumor heterogeneity compared to normal tissue may expand the spectrum of transcribed genes at any given time. In general, while normal cells operate under strict regulatory mechanisms to maintain tissue-specific functions and conserve energy, cancer cells are free to simultaneously utilize a larger fraction of the genome. The cost of an expanded transcriptome is potentially higher immunogenicity, as it can give rise to additional proteins recognizable by the immune system.

### Machine learning approaches

As more single-cell omics datasets become available, constructing machine learning approaches for the identification of malignant cells becomes increasingly feasible. One of the first machine-learning approaches was proposed by van Galen et al. to distinguish malignant and normal cells in AML. The authors started from confident normal bone marrow cells and tumor cells of different cancer subtypes with detected mutations to train a random forest classifier. They then applied the classifier to predict whether the left-out cells without detected mutations in the transcripts were normal or malignant[38]. This approach allowed the authors to obviate the limited sensitivity of SNA calling on scRNA-seq data and complement the classification by random forest predictions based on single-cell gene expression.

scATOMIC is a machine-learning algorithm for pan-cancer classification of cell types in scRNA-seq which also includes a module for discriminating normal from malignant cells[82]. The method is trained on single-cell data from 19 common cancer types and is structured in a cellular hierarchy, where related cell types are placed under the same parent nodes. Cells of a query dataset are classified according to transcriptional similarity to the training set, traversing the hierarchy down to the most fine-grained cell type supported by a minimum confidence score. The predictions include a provisional "cancer cell" class for several cancer types. To distinguish bona fide cancer cells from normal cells of origin, scATOMIC applies a post-classification method based on cancer-signature scoring. Briefly, putative cancer cells are scored against a database of differentially expressed genes between cancer cells of multiple types and matched normal tissues from OncoDB[81]. Based on the gene set scores, cells are grouped by hierarchical clustering and heuristics are applied to decide which clusters represent malignant or normal cells. A recent study suggested combining the output of scATOMIC with CNA detection using SCEVAN and showed that the consensus of the two approaches yielded superior performance[121].

Ikarus is a machine learning pipeline based on logistic regression and network propagation, aimed at distinguishing malignant cells from normal cells at the single-cell level[122]. The method relies on deriving robust tumor and normal gene signatures as the consensus of differentially expressed genes between normal and malignant cells across multiple datasets. The tumor and normal gene set scores are then used to train a logistic regression classifier, which can be applied to new query data by label propagation through a cell-cell graph. The authors provide a tumor gene signature based

on several cancer types, which can be customized by users if the tool is to be used on an unrepresented cancer type. A similar approach was used in a recent publication by Yu et al., who derived differentially expressed genes (DEGs) between malignant and normal epithelial cells for four carcinoma types. The union of the DEGs from multiple datasets was then used to construct a logistic regression classifier, called scMalignantFinder[123]. In their own benchmark, the algorithm outperformed competing methods, in particular copy-number-based classification of malignant cells.

In principle, machine learning methods could be agnostic to the molecular mechanisms driving the differences between normal and malignant cells and be entirely data-driven. Approaches based on deep neural networks are being proposed for the specific task of cancer cell identification[124–126] and are expected to become increasingly prevalent as large scRNA-seq datasets become available. Nevertheless, generating accurate machine learning methods heavily depends on well-annotated training data, for which an understanding of fundamental principles of malignancy remains essential.

## Conclusions

Several transcriptional features have been exploited in research to identify malignant cells from scRNA-seq datasets (summarized in Table 1). Most studies have mainly relied on three features, alone or in combination: *i)* expression of cell-of-origin marker genes (e.g., epithelial markers for carcinomas); *ii)* inter-patient heterogeneity, as measured by patient-specific clustering of cancer cells; and *iii)* detection of copy-number alterations. Depending on the cancer type, however, these three features may not be sufficient to confidently and specifically distinguish malignant cells from normal cells. For cancers with no or infrequent CNAs, for example, detection of single-nucleotide variants may be necessary to capture more subtle differences between normal and malignant cells. Other cancer types may rely on specific chromosomal aberrations such as gene fusions; in this case, specialized tools and pipelines may need to be applied. In cancers that are characterized by specific pathway alterations, gene markers and multi-gene signatures can guide the identification of malignant cells. Enrichment in proliferating cells may also be a useful feature to support classification. Finally, several well-established cancer cell aberrations, including immune evasion mechanisms such as MHC downregulation and PD-L1 over-expression, among others, are potentially detectable in single-cell omics data and could complement the identification of malignant cells. In practical terms, we note that most of these features can be measured by differential activation of specific genes or gene signatures.

Tumors are not only heterogeneous across subjects (*inter-patient*), but they also present internal variability within the same tumor. Recent meta-analyses have identified a relatively small number of recurrent gene meta-programs that explain the majority of variability within tumors across many cancer types[11,114]. Such *intra-tumor* heterogeneity is a consequence of clonal structures within the tumor, as well as the spatial distribution of cancer cells and their interactions with the TME. Intra-tumor heterogeneity can be a complicating factor in the identification of malignant cells, since specific criteria discussed here may only apply to a subset of the cells within a tumor. For example, only a fraction of cells may activate particular pathways such as proliferation at a given time; specific subclones of a tumor may harbor different copy-number variations; and a subset of cancer cells may shed cell-of-origin markers through lineage plasticity programs such as epithelial-to-mesenchymal transition[127]. It can be argued that such "division of labor" by tumor subpopulations provides an evolutionary advantage for cancers to survive treatment regimens and recur months or years after tumor regression[128]. In a broad sense, normal epithelial cells may also be considered as a special sub-clone of a tumor: the sub-clone without copy-number alterations, displaying similar transcriptomics activity across multiple subjects. Approaches for malignant cell identification that aim to be comprehensive will benefit from combining multiple classification criteria in order to cover all the "flavors" of cancer cells within heterogeneous tumors. However, plug-and-play pipelines that integrate multiple features for the identification of malignant cells are currently lacking.

The collective experience of ten years of single-cell analysis indicates that transcriptional processes are often better represented by clusters of cells rather than individual cells. On one hand, this is a consequence of the limited sensitivity of scRNA-seq, with missing detection of transcripts in a significant fraction of cells (a.k.a. dropout). This can partly explain the success of CNAs for cancer cell identification: rather than using expression of individual genes or mutations, CNAs measure the altered expression of multiple adjacent genes along a chromosome, thereby reducing sparsity[128]. On the other hand, the defining hallmarks of cancer could be thought of as properties of groups of cells, rather than of individual cells. Within a tumor, not all cells present the features of malignancy; and conversely, individual cells often display mutations and other abnormalities without being necessarily malignant. It is perhaps for these reasons – both technical and biological – that computational tools seem to have converged to confront malignant cell identification as a problem of clusters of cells, rather than of individual cells.

Machine learning methods that explicitly or implicitly rely on all these transcriptional alterations are increasingly becoming available (Table 2). Provided enough well-annotated data, automated methods have the potential to accurately discriminate normal from malignant transcriptomes through the combination of multiple features. Nevertheless, it is possible that RNA readouts may not always be sufficient to accurately classify malignant cells. Integration of scRNA-seq with other modalities, such as chromatin accessibility or protein abundance and spatial localization, may be beneficial to augment the transcriptome with epigenetic and post-translational mechanisms of malignancy.

## Reporting summary

Further information on research design is available in the Nature Portfolio Reporting Summary linked to this article.

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

## Acknowledgements

This work was supported by the SNSF fellowship 180010 (S.J.C.).

## Author contributions

M.A. and S.J.C. conceptualized the review topic and structure. M.A. performed the literature review and led the writing of the manuscript. S.J.C. and J.G. contributed to manuscript writing. All authors revised the manuscript and approved the final version.

## Competing interests

The authors declare no competing interests.
