## [Transparent Peer Review file · Communications Biology]

Identification of malignant cells in single-cell transcriptomics data

Corresponding Author: Dr Massimo Andreatta

Version 0:

Reviewer comments:

Reviewer #1

(Remarks to the Author)

This manuscript presents a systematic review of computational approaches for distinguishing malignant from non-malignant cells from the perspective of single-cell transcriptomics. The review is structured around three main axes: (1) origin-gene inference, (2) cellular heterogeneity, and (3) copy number alterations (CNAs). The authors provide a relatively comprehensive summary of diverse strategies, including mutation pressure, gene fusions, proliferative potential, and pathway annotation, while also offering insights into future directions involving potentially extensible features. Overall, the manuscript is well-organized and covers a broad spectrum of relevant studies. However, I have several comments and suggestions that may help improve the manuscript:

- 1) Regarding the discussion on inter-patient heterogeneity, have the authors considered the possibility that the observed clustering differences may be partially attributable to technical batch effects? It would be beneficial to clarify whether batch correction strategies were applied. Additionally, I recommend incorporating case studies of robust deep learning models that have demonstrated effective feature extraction at the cancer subtype level (e.g., PMID: 38809990, PMID: 39865982).
- 2) In the section discussing gene fusions, it may be helpful to briefly introduce the fundamental principles underlying sequence alignment algorithms commonly used in fusion detection. This would enhance readers' understanding of how such methods contribute to identifying malignant features.
- 3) While the topics of pathway annotation, tumor angiogenesis (as an additional feature), metabolic characteristics, and origin-gene are conceptually distinct, in practice, their analysis in single-cell transcriptomics often relies on enrichment analyses based on predefined gene sets. This methodological overlap may lead to partially redundant discussions. The authors could consider including more differentiated case studies or providing a comparative analysis to highlight key differences and complementarities among these approaches.
- 4) In the discussion of differentially expressed genes, it may be valuable to incorporate recent developments around the concept of gene ordering, or to reference unsupervised approaches such as non-negative matrix factorization (NMF) that identify gene programs capable of distinguishing malignant cells (e.g., PMID: 38368754).
- 5) In the section on potentially extensible features, I suggest the inclusion of frameworks aimed at identifying systematic signatures that display monotonic changes (e.g., consistently increasing or decreasing expression patterns) along the continuum from normal tissue, inflammation, and precancerous lesions to malignancy. In addition, the manuscript could benefit from discussing the concept of oncofetal similarity, as explored in recent case studies (e.g., PMID: 32946775, PMID: 32209487, PMID: 35999292, PMID: 38368754).
- 6) The current discussion of computational methods largely centers on traditional machine learning. I recommend the inclusion of deep learning approaches, which are particularly well-suited for handling the high redundancy and complexity of single-cell transcriptomic data. These models can extract interpretable and high-resolution features more efficiently and have demonstrated utility in broader malignant contexts, such as metastasis, drug resistance, and prognosis.

Reviewer #2

(Remarks to the Author)

The review is a useful guide to computational methods and prospects for the identification of malignant cells in single cell RNAseq datasets. For each approach, advantages and disadvantages, together with case of application, are thoroughly explained.

I find the manuscript to be well-structured and clear.

I have only a few questions:

1. The authors highlight the problem of epithelial-to-mesenchymal transition downregulating expression of cell-of-origin markers in carcinomas. Are there methods to identify this phenomenon from scRNAseq data and still use its signature to identify malignant cells? If not, do the authors consider this a feasible approach for future prospects?
2. The authors list many different computational tool for the identification of CNAs in scRNAseq datasets. Were the tools ever benchmarked against each other? Can the authors provide more specific insights on performance and case of use for each of the described tool?
3. In a similar way, for inter-patient tumor heterogeneity quantification, is a published tool available?
4. Do the authors find a wrapper tool employing multiple and flexible methodologies for the identification of malignant cell populations in scRNAseq datasets is still lacking?

Finally, I think the manuscript would benefit from the addition of a table listing the mentioned computational tools (categorised by used approach), the corresponding publication, pros and cons, suggested case of use, summary of benchmarking results if available and implementation. This would be a helpful tool to readers approaching the computational analysis evaluated in this review.

Version 1:

Reviewer comments:

Reviewer #1

(Remarks to the Author)

The manuscript has been thoroughly revised, with a clear and coherent overall structure. I recommend it for publication.

Reviewer #2

(Remarks to the Author)

I believe the authors addressed previous comments in an exhaustive manner.

Reply to reviewers' comments for manuscript COMMSBIO-25-4044-T

"Identification of malignant cells in single-cell transcriptomics data: first principles and computational solutions" by M. Andreatta et al.

We would like to extend our gratitude to the editor and reviewers for their valuable comments and suggestions. In the following letter, we address the comments and list all changes to the updated manuscript (highlighted in red font).

Reviewer #1 (Remarks to the Author):

This manuscript presents a systematic review of computational approaches for distinguishing malignant from non-malignant cells from the perspective of single-cell transcriptomics. The review is structured around three main axes: (1) origin-gene inference, (2) cellular heterogeneity, and (3) copy number alterations (CNAs). The authors provide a relatively comprehensive summary of diverse strategies, including mutation pressure, gene fusions, proliferative potential, and pathway annotation, while also offering insights into future directions involving potentially extensible features. Overall, the manuscript is well-organized and covers a broad spectrum of relevant studies. However, I have several comments and suggestions that may help improve the manuscript:

1) Regarding the discussion on inter-patient heterogeneity, have the authors considered the possibility that the observed clustering differences may be partially attributable to technical batch effects? It would be beneficial to clarify whether batch correction strategies were applied. Additionally, I recommend incorporating case studies of robust deep learning models that have demonstrated effective feature extraction at the cancer subtype level (e.g., PMID: 38809990, PMID: 39865982).

Thank you for raising an important point. Batch effects can definitely contribute to apparent inter-patient heterogeneity and can be difficult to distinguish from real biological differences. Most studies we reviewed observed in qualitative terms that cancer cells from different patients/samples tend to form individual, patient-specific clusters, as opposed to other cell types which are more well-mixed across samples. Two of the studies mentioned in the review also provided quantitative measures of inter-patient heterogeneity: Yerly et al. measured cluster mixing by a per-cell-type Local Inverse Simpson's Index (LISI) score, and found that cancer cells (as defined by orthogonal criteria such as pathway dysregulation and CNAs) had consistently lower LISI score compared to normal epithelial cells, stromal cells and immune cell types. Similarly, Chan et al. observed that malignant cells had lower mixing (in terms of Shannon entropy) compared to normal epithelial cells or non-epithelial cells. If these observations were only due to batch effects, one would expect similar inter-patient mixing for cancer and normal cells. We included the following paragraph to highlight the potential confounding role of batch effects:

"Batch effects – technical variations introduced during sample collection and processing, library preparation and sequencing – can be a potential confounding factor when assessing inter-patient heterogeneity. Therefore, the evaluation of inter-patient tumor heterogeneity and its associated metrics depend on the availability of multiple samples from different individuals sequenced in a consistent way (i.e. same technology and protocol)."

Additionally, as suggested by the reviewer we included a reference to RFormer (PMID: 39865982) in the discussion of deep-learning-based approaches:

“Approaches based on deep neural networks are being proposed for the specific task of cancer cell identification^{113–115} and are expected to become increasingly prevalent as large scRNA-seq dataset become available.”

2) In the section discussing gene fusions, it may be helpful to briefly introduce the fundamental principles underlying sequence alignment algorithms commonly used in fusion detection. This would enhance readers' understanding of how such methods contribute to identifying malignant features.

Thank you for the suggestion. We included a brief description of the principles used by sequence alignment algorithms for gene fusion detection:

“Multiple computational tools have been developed for the detection of fusion transcripts in bulk RNA-seq data, such as STAR-fusion⁶⁰ and Arriba⁶¹. When aligning RNA-seq reads to a reference genome, these methods focus on the detection of chimeric (fusion) transcripts, where a single read maps to two distinct genomic loci. Several parameters – such as the number of supporting reads, breakpoint consistency, intergenic distance, and presence in databases of known fusions – are often applied to prioritize and filter candidate fusion events.”

3) While the topics of pathway annotation, tumor angiogenesis (as an additional feature), metabolic characteristics, and origin-gene are conceptually distinct, in practice, their analysis in single-cell transcriptomics often relies on enrichment analyses based on predefined gene sets. This methodological overlap may lead to partially redundant discussions. The authors could consider including more differentiated case studies or providing a comparative analysis to highlight key differences and complementarities among these approaches.

We agree with the reviewer that several features (e.g. sustained proliferation, pathway dysregulation or metabolic signatures) share similar types of readouts and are often evaluated in terms of gene signature activity. However, we decided to thematically structure our review in terms of cancer hallmarks, discussing for each hallmark the transcriptional features that could be quantified from scRNA-seq data. We list the “readout” for each feature in Table 1, and added the following remark to the Conclusions:

“[...] several well-established cancer cell aberrations, including immune evasion mechanisms such as MHC downregulation and PD-L1 overexpression among others, are potentially detectable in single-cell omics data and could complement the identification of malignant cells. In practical terms, we note that most of these features can be measured by differential activation of specific genes or gene signatures.”

While gene signatures are the simplest and most common readout for evaluating feature alterations, we recognize that more complex, specialized algorithms have been proposed for specific tasks. For example, machine-learning methods have been developed to predict cell cycling phases, and constraint-based models have been proposed specifically for metabolic states. We added the following two paragraphs to highlight these alternative approaches:

“While cell proliferation is mostly quantified in terms of gene-set scoring for cell cycling phases, machine-learning methods trained specifically for the task of cell cycle-phase prediction have also been proposed (reviewed in ref⁶⁸). Machine-learning methods have the ability to learn complex, multivariate relationships between genes and may provide more robust cell cycle-phase predictions.”

“Metabolic reprogramming can be quantified in terms of gene-set activity and pathway enrichment, or by using more complex constraint-based or kinetic modeling for the reconstruction of metabolic fluxes (reviewed in ref ¹⁰²). Regardless of the algorithm employed, differences in metabolic activity may be exploited to distinguish normal from malignant cells in scRNA-seq.”

4) In the discussion of differentially expressed genes, it may be valuable to incorporate recent developments around the concept of gene ordering, or to reference unsupervised approaches such as non-negative matrix factorization (NMF) that identify gene programs capable of distinguishing malignant cells (e.g., PMID: 38368754).

Unsupervised approaches for gene program extraction (such as NMF) are useful tools to characterize the main axes of transcriptional variation of scRNA-seq data. These methods have been successfully applied to characterize intra-tumoral heterogeneity in terms of a small set of interpretable meta-programs (e.g. Gavish et al. 2023, Tyler et al. 2025). While malignant-cell specific programs could be learned “de novo” from unsupervised analyses, in our view they should also be captured by a combination of the other features discussed in this review. We highlight some studies that rely on NMF for metaprogram discovery in the updated text:

“Tumors are not only heterogeneous across subjects (*inter-patient*), but they also present internal variability within the same tumor. Recent meta-analyses have identified a relatively small number of recurrent gene meta-programs that explain the majority of variability within tumors across many cancer types ^{11,118}.”

We also refer to the new section on epithelial-to-mesenchymal transition (see below), where these NMF approaches were used to extract EMT signatures with potential for discriminating malignant from normal epithelial cells.

5) In the section on potentially extensible features, I suggest the inclusion of frameworks aimed at identifying systematic signatures that display monotonic changes (e.g., consistently increasing or decreasing expression patterns) along the continuum from normal tissue, inflammation, and precancerous lesions to malignancy. In addition, the manuscript could benefit from discussing the concept of oncofetal similarity, as explored in recent case studies (e.g., PMID: 32946775, PMID: 32209487, PMID: 35999292, PMID: 38368754).

Thank you for this excellent suggestion. Re-expression of fetal-like cell states by tumor cells (“oncofetal reprogramming”) could be considered as a developmental mimicry axis that supports the acquisition of cancer hallmarks, such as sustained proliferative signaling, replicative immortality and phenotypic plasticity. Specific genes associated with oncofetal reprogramming (e.g. AFP, SALL or CEA) could potentially be used to distinguish normal from malignant cells. We included a new section that discusses oncofetal reprogramming as a new potential feature for malignant cell identification:

Oncofetal reprogramming

Re-expression of fetal-like cell states allows cancer cells to mimic characteristics of embryonic or fetal development, adopting gene expression patterns and behaviors typical of early developmental stages. By reactivating developmental signaling pathways and cellular states that are dormant or inactive in normal adult tissues, oncofetal reprogramming promotes tumor development and invasion, phenotypic plasticity and evasion of immune recognition ¹¹³. Abnormal expression of fetal antigens in tumors has been reported in several cancer types and has been used as a clinical biomarker. For example, re-

expression of the oncofetal proteins *SALL4* and *AFP* is often observed in hepatocellular carcinoma^{114,115}. High expression of *IGF2BP* family proteins has been reported across cancers of multiple tissues, including colon, liver, kidney, pancreas, and female reproductive organs¹¹⁶. Other common genes with oncofetal properties include *CEA*, *CA125*, *LIN28B*, *H19* and *FOXM1*¹¹³. Using scRNA-seq data, a recent study of cervical cancer found that oncofetal protein *SALL4* was significantly upregulated in cancerous cells compared to normal epithelial cells¹¹⁷. Because of their specific expression in malignant cells, oncofetal signatures can be potentially useful to identify cancer cells in single-cell data.

Additionally, we also added a section on EMT, which can be considered as a continuous variable with intermediate states from epithelial to mesenchymal.

Drivers of invasion

Cancer invasion refers to the ability of tumor cells to break through their local tissue boundaries, penetrate the surrounding extracellular matrix, and invade nearby tissues. Epithelial-to-mesenchymal transition (EMT) is a fundamental process by which cancer cells lose their adhesion and polarity, adopting mesenchymal traits that increase their migratory and invasive capacities. These morphological and functional changes are accompanied by the expression of transcriptional programs characterized by gradual loss of epithelial markers (e.g. *EPCAM* and *CDH1*) while gaining mesenchymal characteristics (e.g. *VIM* and *FN1*)¹¹¹. Large meta-analyses of scRNA-seq have identified recurrent EMT programs across multiple cancer types, providing signatures that could aid the identification of malignant cells^{11,112}. While EMT can be a useful feature to distinguish malignant from normal cells, it has limitations. EMT is a continuous, dynamic and reversible process, which gives rise to partial and intermediate states that affect only a fraction of cancer cells in a tumor.

To accommodate these additions, we modified Figure 1 with one new hallmark (“Activating invasion”) and two new transcriptional readouts (“EMT signatures” and “oncofetal reprogramming”).

6) The current discussion of computational methods largely centers on traditional machine learning. I recommend the inclusion of deep learning approaches, which are particularly well-suited for handling the high redundancy and complexity of single-cell transcriptomic data. These models can extract interpretable and high-resolution features more efficiently and have demonstrated utility in broader malignant contexts, such as metastasis, drug resistance, and prognosis.

Thank you for your suggestion. In the section “machine learning approaches” we reviewed the most

prevalent automated methods for the identification of malignant cells in scRNA-seq. These include random forests (scATOMIC) and logistic regression (Ikarus, scMalignantFinder), but also deep-learning methods (TCfinder, Cancer-Finder and RFormer). We agree that deep-learning methods can be expected to become a dominant approach to distinguish malignant from normal cells; however, at this time they are not the standard practice for cancer cell identification. As suggested by the reviewer we included a reference to RFormer (PMID: 39865982) in the discussion of deep-learning-based approaches:

“Approaches based on deep neural networks are being proposed for the specific task of cancer cell identification ^{113–115} and are expected to become increasingly prevalent as large scRNA-seq dataset become available.”

Reviewer #2 (Remarks to the Author):

The review is a useful guide to computational methods and prospects for the identification of malignant cells in single cell RNAseq datasets. For each approach, advantages and disadvantages, together with case of application, are thoroughly explained. I find the manuscript to be well-structured and clear.

Thank you for the positive evaluation of our work.

I have only a few questions:

1. The authors highlight the problem of epithelial-to-mesenchymal transition downregulating expression of cell-of-origin markers in carcinomas. Are there methods to identify this phenomenon from scRNAseq data and still use its signature to identify malignant cells? If not, do the authors consider this a feasible approach for future prospects?

Thank you for giving us the opportunity to expand on this point. Indeed, EMT is a fundamental process for cancer cells to acquire invasive and metastatic potential – a key hallmark of cancer. EMT does indeed involve loss of cell-of-origin markers, but at the same time the acquisition of mesenchymal features, giving rise to intermediate phenotypes. Therefore, signatures of EMT could be potentially exploited for cancer cell identification. We decided to include a new section in the future prospects to discuss these points, and modified Figure 1 to include “Activation of invasion” as a cancer hallmark, with EMT as its main transcriptional readout.

Drivers of invasion

Cancer invasion refers to the ability of tumor cells to break through their local tissue boundaries, penetrate the surrounding extracellular matrix, and invade nearby tissues. Epithelial-to-mesenchymal transition (EMT) is a fundamental process by which cancer cells lose their adhesion and polarity, adopting mesenchymal traits that increase their migratory and invasive capacities. These morphological and functional changes are accompanied by the expression of transcriptional programs characterized by gradual loss of epithelial markers (e.g. *EPCAM* and *CDH1*) while gaining mesenchymal characteristics (e.g. *VIM* and *FN1*) ¹¹¹. Large meta-analyses of scRNA-seq have identified recurrent EMT programs across multiple cancer types, providing signatures that could aid the identification of malignant cells ^{11,112}. While EMT can be a useful feature to distinguish malignant from normal cells, it has limitations. EMT is a continuous, dynamic and reversible process, which gives rise to partial and intermediate states that affect only a fraction of cancer cells in a tumor.

2. The authors list many different computational tool for the identification of CNAs in scRNAseq datasets. Were the tools ever benchmarked against each other? Can the authors provide more specific insights on performance and case of use for each of the described tool?

Independent benchmarks of the performance of CNV methods are scarce, especially for the task of malignant cell identification. However, in the past few months two benchmarks have tackled this problem, and found that methods that exploit allelic shift signals (namely Numbat and CaSpER) were the top performers for identification of CNAs. When Fastq files are not available, both benchmarks agree that CopyKAT is the recommended choice. We added this information to the manuscript:

“Recent benchmarks have found that methods that exploit allelic shift signals (such as Numbat and CaSpER) have superior performance for CNA identification; when only expression matrices are available, CopyKAT is the recommended method ^{28,29}.”

3. In a similar way, for inter-patient tumor heterogeneity quantification, is a published tool available?

Inter-patient heterogeneity can be quantified by metrics of patient mixing. For example, Yerly et al. measured cluster mixing by a per-cell-type LISI score (Korsunsky et al. 2019). Code implementations for this metric are available (e.g. <https://github.com/immunogenomics/LISI> and <https://github.com/carmonalab/scIntegrationMetrics>). We added these resources in the new Table 2 (see below), according to the last suggestion by this reviewer.

4. Do the authors find a wrapper tool employing multiple and flexible methodologies for the identification of malignant cell populations in scRNAseq datasets is still lacking?

Indeed, comprehensive pipelines/wrappers that integrate multiple features for malignant cell identification are still lacking. The most automated methods that can be used out of the box are those discussed in the “machine learning approaches” section. For example, scATOMIC returns cell type annotations including malignant vs. normal predictions. It also internally implements CNA inference using CopyKAT, thereby providing multiple readouts for the prediction of cancer cells. We added the following sentence to the conclusions, to reflect on the lack of multi-feature pipelines:

“Approaches for malignant cell identification that aim at being comprehensive will benefit from combining multiple classification criteria in order to cover all the “flavors” of cancer cells within heterogeneous tumors. However, plug-and-play pipelines that integrate multiple features for the identification of malignant cells are currently lacking.”

Finally, I think the manuscript would benefit from the addition of a table listing the mentioned computational tools (categorised by used approach), the corresponding publication, pros and cons, suggested case of use, summary of benchmarking results if available and implementation. This would be a helpful tool to readers approaching the computational analysis evaluated in this review.

This is a good idea. We added a new table (Table 2) listing selected resources available for the identification of malignant cells.

Table 2: Selected tools and resources for the identification of malignant cells in scRNA-seq data.

Resource	Type/readout	Comments	Availability and references
InferCNV	Copy number alterations	Arguably the most widely used method for CNA detection in scRNA-seq	https://github.com/broadinstitute/infercnv ²³
CopyKAT		Among top performers in recent benchmarks, especially when using only gene expression matrix	https://github.com/navinlabcode/copykat ²⁴
Numbat		Exploits allelic imbalance to improve CNA prediction; requires sequencing reads	https://github.com/kharchenkolab/numbat ²⁷
LISI	Inter-patient heterogeneity	A simple metric of patient mixing	https://github.com/immunogenomics/LISI ⁴⁵
scIntegrationMetrics		Implements LISI and additional metrics	https://github.com/carmonalab/scIntegrationMetrics ¹²⁹
scAllele	Single nucleotide alterations	SNA detection tailored for scRNA-seq	https://github.com/qxiaolab/scAllele ⁵⁰
Monopogen		SNA calling (germline + somatic) leveraging linkage disequilibrium from reference panels	https://github.com/KChen-lab/Monopogen ⁵¹
STAR-fusion	Fusion transcripts	Primarily designed for bulk RNA-seq, but can be adapted for single-cell data	https://github.com/STAR-Fusion/STAR-Fusion ⁶²
scFusion		Specific for gene fusion detection at single-cell resolution	https://github.com/XiDsLab/scFusion ⁶⁵
UCell	Gene signature scoring	Simple and intuitive rank-based gene set scoring	https://github.com/carmonalab/UCell ¹³⁰
GSVA		Implements methods for gene set enrichment analysis	https://github.com/rcastelo/GSVA ¹³¹
scATOMIC	Automated classifier	Integrated pipeline for cell type classification, including malignant vs. normal cells	https://github.com/abelson-lab/scATOMIC ⁸²
Ikarus		Relies on DEG signatures between normal and malignant cells	https://github.com/BIMSBbioinfo/ikarus ¹²²
scMalignantFinder		Uses logistic regression trained on curated pan-cancer gene signatures and DEGs	https://github.com/Jonyyqn/scMalignantFinder ¹²³
OncoDB	Database	Collates expression profiles for cancer vs. normal tissues	https://oncodb.org/ ⁸¹
3CA		Provides robust transcriptional meta-programs for several cancer types	https://www.weizmann.ac.il/sites/3CA/ ¹¹⁴
HPA		Includes scRNA-seq expression profiles for many tissues and cell types	https://www.proteinatlas.org/ ¹³²